# Disentangling acute motor deficits and adaptive responses evoked by the loss of cerebellar output

Nirvik Sinha[1,2,3]*, Sharon Israely[1], Ora Ben Harosh[1], Ran Harel[4],
Julius PA Dewald[2,3,5,6], Yifat Prut[1]*

[1]Edmond and Lily Safra Center for Brain Sciences, The Hebrew University of
Jerusalem, Jerusalem, Israel; [2]Department of Physical Therapy and Human Movement
Sciences, Northwestern University Feinberg School of Medicine, Chicago, United
States; [3]Interdepartmental Neuroscience Center, Northwestern University Feinberg
School of Medicine, Chicago, United States; [4]Department of Neurosurgery, Sheba
Medical Center, Tel-Hashomer Affiliated to Tel Aviv University, Israel; [5]Department
of Biomedical Engineering, Northwestern University, Evanston, United States;
[6]Department of Physical Medicine & Rehabilitation, Northwestern University
Feinberg School of Medicine, Chicago, United States

*For correspondence:
Nirvik.Sinha@mail.huji.ac.il (NS);
yifat.prut@mail.huji.ac.il (YP)

Reviewing Editor: J Andrew
Pruszynski, Western University,
Canada

## eLife Assessment

Using a unique cerebellar disruption approach in non-human primates, this study provides **valuable**
new insight into how cerebellar inputs to the motor cortex contribute to reaching. The findings
**convincingly** demonstrate that reaching movements following cerebellar disruption slow down
because of both an acute deficit in producing muscle activity as well as a progressive decline in
compensating for limb dynamics. This work will be of interest to neuroscientists and clinicians inter-
ested in cerebellar function and pathology.

**Abstract** Patients with cerebellar damage experience various motor impairments, but the
specific sequence of primary and compensatory processes that contribute to these deficits remains
unclear. To clarify this, we reversibly blocked cerebellar outflow in monkeys engaged in planar
reaching tasks. This intervention led to a spatially selective reduction in hand velocity, primarily due
to decreased muscle torque, especially in movements requiring high inter-joint torque coupling.
When examining repeated reaches to the same target, we found that the reduced velocity resulted
from both an immediate deficit and a gradually developing compensatory slowing to reduce passive
inter-joint interactions. However, the slowed hand velocity did not account for the fragmented and
variable movement trajectories observed during the cerebellar block. Our findings indicate that
cerebellar impairment results in motor deficits due to both inadequate muscle torque and an altered
motor control strategy for managing impaired limb dynamics. Additionally, impaired motor control
elevates noise, which cannot be entirely mitigated through compensatory strategies.

## Introduction

The ability to coordinate well-timed movements greatly relies on the cerebellum as evident from
studies of individuals with cerebellar ataxia (*Beppu et al., 1984*). The cerebellum regulates the timing
of movements by predicting the mechanical interactions between adjacent joints (*Ivry and Keele,*

*1989*; *Manto et al., 2012*), a computation that requires an internal model of feedforward control (*Shidara et al., 1993*; *Shadmehr and Mussa-Ivaldi, 1994*; *Wolpert et al., 1995*; *Wolpert et al., 1998*; *Kawato, 1999*; *Sainburg et al., 1999*; *Popa et al., 2013*). However, motor impairments exhibited by individuals with cerebellar deficits extend much beyond poor motor timing. For example, during the acute phase of cerebellar injury, there is a loss of muscle tone and weakness of voluntary movements (*Holmes, 1917*; *Goldstein, 1927*; *Holmes, 1939*). Cerebellar patients also exhibit movement-related sensory deficits such as an impaired visual perception of stimulus movement and perception of the limb position sense (i.e. proprioception or state estimation) during active movements (*Therrien and Bastian, 2015*; *Weeks et al., 2017*).

Clinical studies of cerebellar patients have yielded conflicting insights into the pathophysiology caused by a loss of cerebellar function. Some researchers suggest that motor impairments stem primarily from an inability to compensate for inter-joint interactions (*Bastian et al., 1996*; *Bastian et al., 2000*), while others argue that insufficient muscle torque generation is the central issue (*Topka et al., 1998a*; *Boose et al., 1999*). These discrepancies likely arise from the variability among individuals with chronic cerebellar deficits, where primary and compensatory effects are intertwined and difficult to separate. Moreover, individuals with cerebellar lesions often have differences in lesion etiology and associated damage beyond the cerebellum itself. As a result, the specific sequence of events that leads to motor impairments following the loss of cerebellar signals remains unclear.

To investigate this issue, we trained monkeys to perform a planar reaching task with arm support provided by an exoskeleton. We then reversibly disrupted cerebellar communication with other neural structures using high-frequency stimulation (HFS) of the superior cerebellar peduncle, assessing the impact of this perturbation on subsequent movements. Although our approach primarily affects cerebellar output to the motor cortex, it also disrupts fibers carrying input signals (e.g. spinocerebellar) and pathways to various subcortical targets (e.g. cerebello-rubrospinal). Thus, our manipulation broadly interferes with cerebellar communication. Previously, we have shown that HFS can effectively and reversibly interfere with the normal outflow of cerebellar signals (*Nashef et al., 2019*) since this perturbation accurately replicated the behavioral deficits identified in cerebellar patients (*Bastian, 1997*) and the changes in motor cortical activity that appear after dentate cooling in monkeys (*Meyer-Lohmann et al., 1975*). In the present study, we found that the loss of cerebellar outflow led to reduced hand velocity during reaching movements, reflecting two parallel underlying mechanisms. First, there

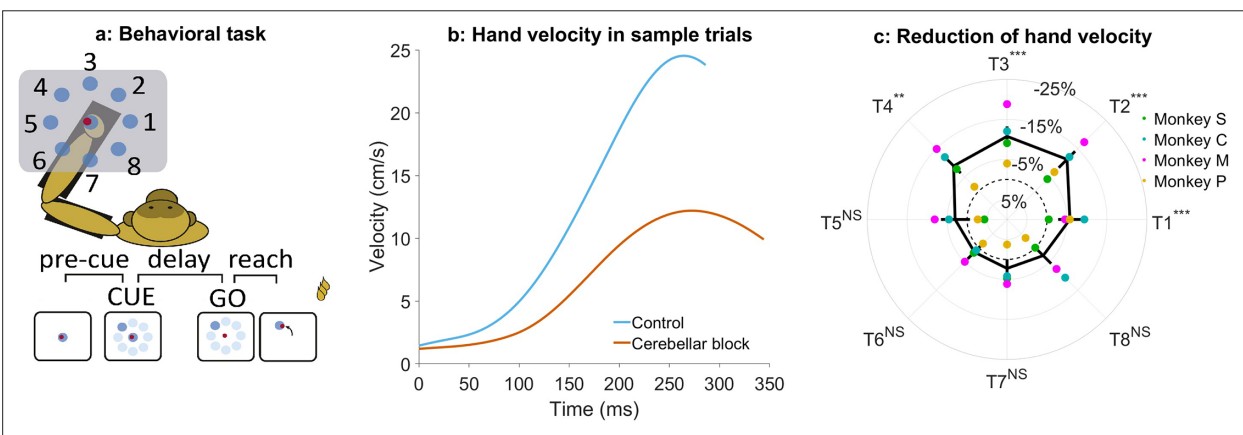

**Figure 1.** Effect of cerebellar block on peak hand velocity. (**a**) Schematic of the monkey performing the delayed reaching task in the KINARM exoskeleton to eight peripheral targets on the horizontal plane. The locations of the eight targets are numbered. (**b**) Movement-onset aligned hand velocity profiles during a sample control and cerebellar block trial performed by monkey C to target 2. Entry of the control cursor into the peripheral target marked the end of the movement for the trial. Since the monkeys did not have to stop their movements fully for the trial to end (see Methods for a detailed explanation of this), the traces appear cutoff at the beginning of the deceleration/stopping phase of the movement. (**c**) Target-wise effect of the cerebellar block on peak hand velocity. For each session, the target-wise reduction in the median peak hand velocity during the cerebellar block trials was computed relative to that of control trials. The depicted values are the means ± 95% confidence intervals across all sessions pooled from all four monkeys. The means of individual monkeys are overlaid. The dashed circle indicates no change. Statistical significance is denoted as follows: p≥0.05 NS, p<0.05*, p<0.01**, p<0.001***. [T1-8: Targets 1–8].

The online version of this article includes the following figure supplement(s) for figure 1:

**Figure supplement 1.** Effect of cerebellar block on peak hand velocity during within-trial vs. inter-trial movements.

was a reduction in muscle torques, even for movement directions where inter-joint interactions were low. Second, the reduction in hand velocity exhibited a spatial tuning and was greater in directions that generated large inter-joint interactions, thereby reflecting an inability to compensate for limb dynamics. The time course of these two mechanisms was identified by analyzing the sequences of movements made to the same target during the cerebellar block. During such movements, hand velocity was low initially, reflecting a primary muscle torque deficit, and declined further in successive trials, revealing an adaptive motor strategy to minimize inter-joint interactions. Finally, we found that the decrease in velocity during cerebellar block could not fully explain the noisy and decomposed movements observed under these conditions, which may be due to an impaired state estimate of the limb that leads to the faulty control of inter-joint interactions. Taken together, we show that cerebellar output is essential for generating sufficient muscle torques to allow for fast and stereotypical movements while overcoming the complex inter-joint interactions inherent in such actions. The loss of this output triggers a sequence of early primary effects followed by compensatory responses aimed at mitigating these impairments. Together, these processes result in significantly slower movements.

## Results

### Slower reaching movements in cerebellar block are due to weakness and impaired limb dynamics

We trained four adult female monkeys (Macaca fascicularis) to sit in a primate chair with their upper limb supported by an exoskeleton (KINARM system) and perform planar center-out movements (*Figure 1a*). The spatial location of the targets enabled us to dissociate between shoulder-dominant (lateral targets) vs. shoulder-elbow movements (diagonal and straight targets, respectively). Each trial began with the monkey holding a cursor within a central target. After 500–800 ms, one of eight peripheral targets appeared (cue). Following a 450–850 ms delay, the central target disappeared ('go' signal), prompting the monkey to reach the peripheral target within 500–900 ms and briefly hold the cursor there (100–150 ms) to receive a reward. Between successive trials, monkeys repositioned the cursor back to the central target. The experiment was conducted across several sessions (i.e. days). Each session was divided into 3–4 sub-sessions. Each sub-session comprised a block of control trials (~80) followed by a block of trials (~50) with high-frequency stimulation (130 Hz) of the superior cerebellar peduncle.

Cerebellar block significantly reduced the ability of the monkeys to perform the task as measured by the success rate across all the recorded sessions (mean success rate in control = 86.5%, CI [85.8, 87.2] vs. cerebellar block = 75.3%, CI [73.9, 76.7], p<0.001; *Supplementary file 1*). Since the monkeys had to meet strict timing criteria to perform the movements successfully during the task, we began by examining the hand kinematics. *Figure 1b* shows profiles of hand velocity in sample trials collected during the control (blue) and cerebellar block (vermillion) conditions. This example shows that movements exhibited a significant reduction in the peak hand velocity during cerebellar block. However, when compared across the targets, we found that the reduction in hand velocity was strongly modulated by the target direction (one-way ANOVA of the change in peak hand velocity, the effect of cerebellar block: p=0.01, the effect of targets: p<0.001; *Supplementary file 2*). *Post hoc* comparisons further revealed that the effect of cerebellar block on outward reaching movements (targets 1–4) i.e., movements involving shoulder flexion was significantly higher than on inward reaching movements (targets 5–8) i.e., movements involving shoulder extension (*Figure 1c*; mean decrease of 9.7%, CI [8.2, 11.2], p<0.001 for targets 1–4 compared to mean decrease of 2.5%, CI [0.9, 4.0], p=0.21 for targets 5–8) even though there was no difference in the peak hand velocity between the movements to the two target groups during control (p=0.88, *Supplementary file 3*).

Previous studies have demonstrated that Purkinje neurons of the cerebellar cortex exhibit weaker modulation during task-irrelevant movements compared to task-related movements (*Pi et al., 2024*; *Hage et al., 2025*). Therefore, we examined whether the impact of the cerebellar block was limited specifically to target-directed, reward-associated movements or if it generalized to all movements performed during the task. To this end, we compared peak hand velocities between target-directed (within-trial, reward-relevant) movements and subsequent return-to-center (out-of-trial) movements. Our analysis revealed that cerebellar block resulted in a significantly larger reduction in peak velocity for target-directed movements compared to return-to-center movements (7.5%, CI [6.5, 9.0] vs. 1.7%,

CI [–1.8, 3.0]; p<0.001; *Figure 1—figure supplement 1*). These findings suggest that the cerebellum plays a preferential role in optimizing reward-associated, within-trial movements rather than equally influencing all movements.

In the task used in our study, hand movements were made by controlling the muscle torque at the shoulder and elbow joints (*Scott, 2004*). However, simultaneous movements of both joints also generated coupling torques due to passive interactions between limb inertial properties and joint velocities (*Bernshteĭn, 1967*, *Hollerbach and Flash, 1982*). To identify the target-specific muscle and coupling torques, we employed an inverse dynamics model of the upper limb (*Fagg et al., 2009*) which enabled us to distinguish between the different contributions to the net torque experienced by each joint. According to this model, the net torque at each joint is the sum of both the active muscle torque and the passive coupling torque acting at that joint (see Methods section for details).

Using the inverse dynamics model, we found that the target-specific reduction in hand velocities was produced by changes in muscle commands (e.g. *Figure 2a and b*). Specifically, we observed that the muscle torque impulses calculated by integrating the torque profiles over the early phase of movements when acceleration was positive, were reduced in a target-specific manner during the cerebellar block (*Figure 2c*; *Supplementary file 4* and *Supplementary file 5*). This reduction was particularly pronounced at the shoulder joint. Furthermore, as in the case of hand velocities, *post hoc* comparisons revealed that the shoulder joint experienced a significantly greater reduction in muscle torque for targets 1–4 vs. 5–8 (*Figure 2d*; mean decrease of 10.6%, CI [9.0, 11.2], p<0.001 for targets 1–4 compared to mean decrease of 3.3%, CI [–1.3, 5.2] p=0.21 for targets 5–8). In contrast, the slight reduction in elbow torques due to the cerebellar block did not reach significance for any of the targets (*Figure 2e*). Collectively, these findings indicate that during the cerebellar block, there is a reduction in shoulder muscle torque specifically for targets involved in outward-reaching movements, which leads to the target-specific reduction in hand velocity.

Several studies have shown that individuals with cerebellar deficits exhibit an inability to correctly compensate for inter-joint interactions (*Bastian et al., 1996*; *Bastian et al., 2000*). Therefore, we asked whether the reduction in muscle torques and hand velocities due to the cerebellar block could be associated with the coupling torques generated during movements in various directions. *Figure 3a and b* show the coupling torque profiles at the shoulder and elbow joints, respectively, during sample trials. We observed that the coupling torque impulses varied in a target-dependent manner during both the control and cerebellar block conditions (*Figure 3c*). Furthermore, the reduction in peak hand velocities during cerebellar block was correlated to the net coupling torque impulse (calculated as the sum of absolute coupling torque impulses at both joints) during control, particularly for the outward reaching movements (*Figure 3d*; targets 1–4: partial correlation coefficient to account for the variability across monkeys, ρ=0.32, CI [0.23, 0.41], p<0.001). This indicated that the effect of the cerebellar block on hand velocity during the outward reaching movements was strongest for targets with high coupling torques. Nevertheless, despite the significant correlation between the reduced hand velocity and coupling torques, cerebellar block reduced the hand velocity even in outward reaches where the overall coupling torque was low i.e., movements to target 1, mean reduction in hand velocity = 5.8%, CI [2.8, 8.7], p<0.001 (movements to all outward targets involved both shoulder and elbow joints, but the contribution of each joint varied in a target-dependent manner. Specifically, movements to targets 1 exhibited negligible net torque at the elbow, meaning the elbow remained largely stationary during these movements as shown in *Figure 2—figure supplement 1*. Consequently, the shoulder experienced correspondingly low coupling torques as shown in *Figure 3c*). This finding indicated a reduction in the motor command which occurred in addition and beyond the effect attributable to the impaired control of coupling torques alone.

Overall, the cerebellar block produced a significant impact on the kinematics and dynamics of outward reaching movements. Specifically, the cerebellar block caused slower hand velocities due to a systematic, target- and joint-specific reduction in muscle torques for these movements, which strongly correlated with the coupling torques. Additionally, a reduction in muscle torques was also observed during the outward reaching movements with low coupling torques, indicating an additional component of muscle torque insufficiency that further exacerbated motor impairment following cerebellar block.

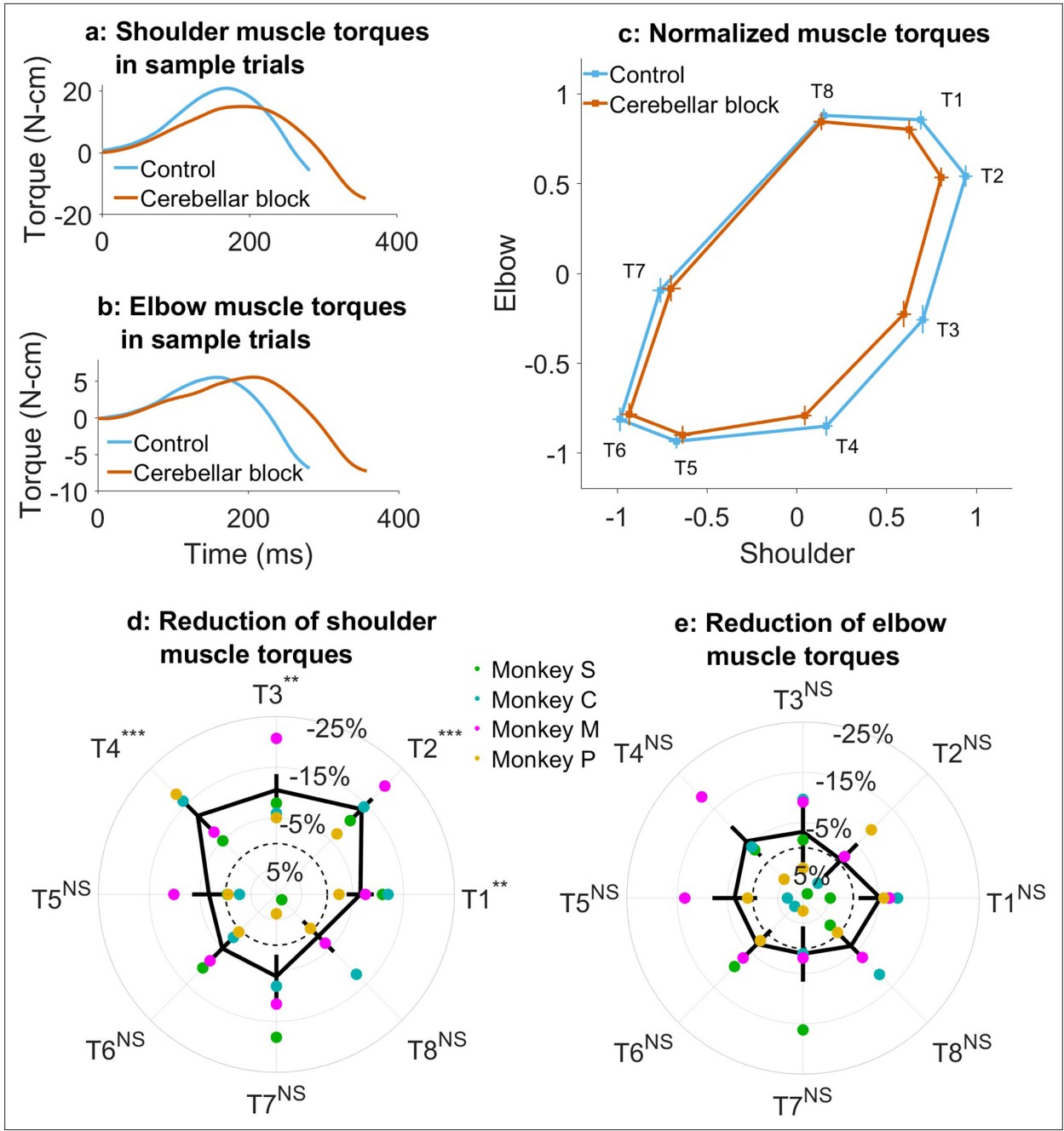

**Figure 2.** Effect of cerebellar block on muscle torques. Movement-onset aligned shoulder (**a**) and elbow (**b**) muscle torque profiles during a sample control and cerebellar block trial performed by monkey C to target 2. Entry of the control cursor into the peripheral target marked the end of the movement for the trial. Since the monkeys did not have to stop their movements fully for the trial to end (see Methods for a detailed explanation of this), the traces appear cutoff at the beginning of the deceleration/stopping phase of the movement. (**c**) Normalized muscle torque impulse at the shoulder vs. elbow per target during control and cerebellar block. For each session, we computed the target-wise median muscle torque impulse during the acceleration phase of the movement across the control and cerebellar block trials. They were then normalized by the maximum absolute torque impulse across all the targets. The depicted values are the means across all the sessions pooled from the data of all four monkeys. The 95% confidence interval of the means is indicated by the horizontal and vertical bars for the shoulder and elbow joints, respectively. Target-wise effect of cerebellar block on shoulder (**d**) and elbow (**e**) muscle torque impulse. For each session, the target-wise reduction in the median torque impulse during the cerebellar block trials was computed relative to the maximum absolute value of the target-wise medians observed during the control trials. The depicted values are the means ± 95% confidence intervals across all sessions pooled from all four monkeys. The means of individual monkeys are overlaid. The dashed circle indicates no change. Statistical significance is denoted as follows: $p \geq 0.05$ NS, $p < 0.05$*, $p < 0.01$**, $p < 0.001$***. [T1-8: Targets 1–8].

The online version of this article includes the following figure supplement(s) for figure 2:

**Figure supplement 1.** Normalized net torque impulse at the shoulder vs. elbow per target during control and cerebellar block conditions.

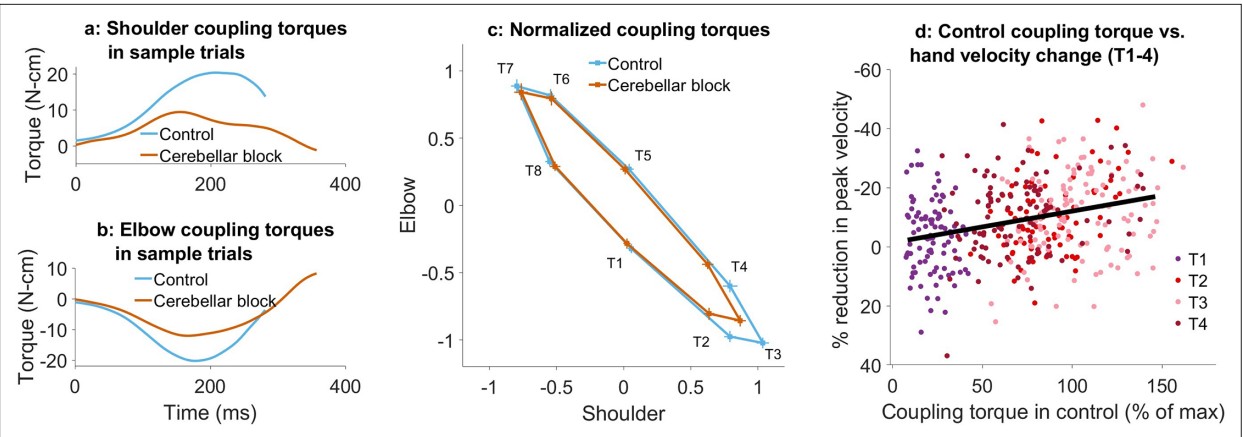

**Figure 3.** Effect of cerebellar block on coupling torques. Movement-onset aligned shoulder (**a**) and elbow (**b**) coupling torque profiles during a sample control and cerebellar block trial performed by monkey C to target 2. Entry of the control cursor into the peripheral target marked the end of the trial. (**c**) Normalized coupling torque impulse at the shoulder vs. elbow per target during control and cerebellar block. For each session, we computed the target-wise median coupling torque impulse during the acceleration phase of the movement across the control and cerebellar block trials. They were then normalized by the maximum absolute torque impulse across all the targets. The depicted values are the means across all the sessions pooled from the data of all four monkeys. The 95% confidence interval of the means is indicated by the horizontal and vertical bars for the shoulder and elbow joints, respectively. (**d**) The effect of the cerebellar block on peak hand velocity (y-axis) is plotted against the total coupling torque experienced across both joints during control (x-axis) for the outward reaches (T1-4). The effect of the cerebellar block on peak hand velocity was measured by the target-wise change in the median peak hand velocity during the cerebellar block trials relative to that of control trials per session (negative percentage values indicate a stronger reduction due to the cerebellar block). This measure was then correlated to the target-wise median coupling torque during the control trials of the corresponding sessions. The median coupling torque was taken as the sum of the absolute coupling torque impulse at both the shoulder and elbow joints per trial. These values were then normalized by the median across all trials of all sessions per monkey (to yield percentage units like the y-axis). The depicted data contains all the sessions pooled from all four monkeys with individual points representing each of the four targets (color-coded) per session. The black line overlaid on the scatter indicates the least-squares linear regression fit to the data points. [T1-8: Targets 1–8].

## Hand velocity exhibits an adaptive trend during repeated movements to the same target during cerebellar block

The advantage of using a reversible cerebellar block in an animal model is the ability to track the temporal profile of motor impairments induced by this condition, allowing us to distinguish between primary and compensatory effects. To this end, we assessed changes in hand kinematics produced by the cerebellar block across successive trials. Specifically, we examined the evolution of peak hand velocities for outward reaches with low (target 1) vs. high coupling torques (data pooled for movements to targets 2–4) during blocks of control and cerebellar block trials. *Figure 4a* illustrates this procedure. Since the targets were presented in a pseudorandom order during each block of trials, we collected for each block the tested target, while keeping its order of presentation to generate a temporally accurate sequence of trials, all directed towards the same single target.

We found that during the control condition, peak hand velocity remained stable across successive trials. In contrast, cerebellar block affected hand velocity in a trial- and target-dependent manner (*Figure 4b and c*). For movements to target 1, hand velocity was lower than control but unaffected by the temporal sequence of target presentation (trial type ×trial sequence interaction effect: p=0.69, *Supplementary file 6*). However, for movements to targets 2–4, hand velocity was lower than control in the early trials and declined further with successive trials (trial type ×trial sequence interaction effect: p=0.007, *Supplementary file 7*).

To further quantify the effect of trial sequence on movements to targets 2–4, we performed *post hoc* comparisons (*Figure 4d*). These comparisons revealed that during the cerebellar block, hand velocity was 4.3% (CI [1.0, 7.3], p<0.001) lower than control during the first 1–2 trials. With successive trials, the hand velocity declined further before reaching an asymptote during the late trials 11–20 at 8.2% (CI [5.7, 10.7], p<0.001) lower than the control.

Taken together, these results reveal that the hand velocity of outward-reaching movements with high coupling torques was already low initially during cerebellar block, reflecting a primary deficit.

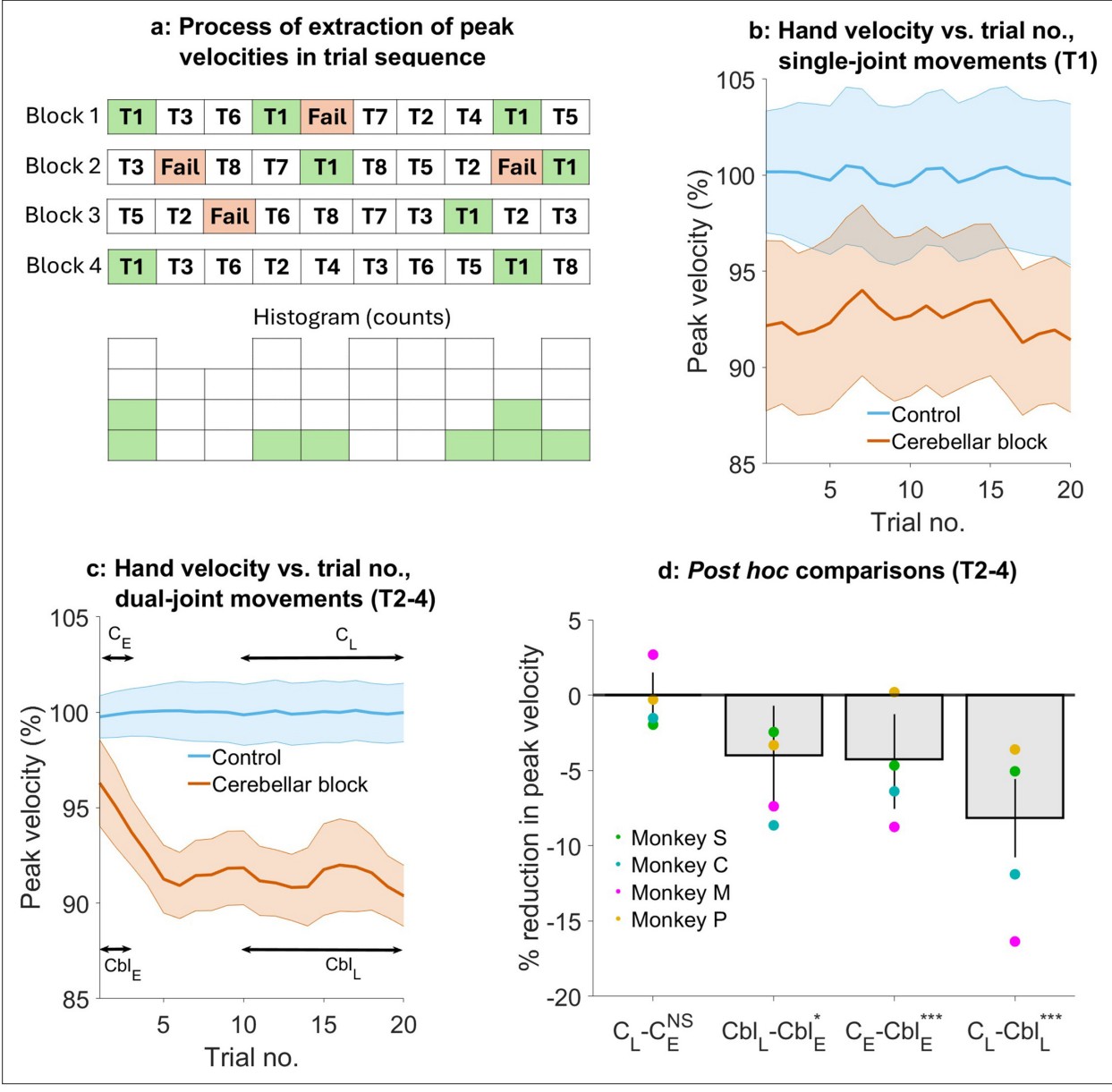

**Figure 4.** Effect of the cerebellar block on hand velocity across successive trials. (**a**) Schematic of the process of extraction of peak hand velocity for movements to a particular target from blocks of control/cerebellar block trials while preserving the order in which the target was presented. For each block, the peak velocity for movements to a particular target (in this example, target 1) was extracted while retaining the trial numbers in which it was presented in the block. This enabled us to examine the evolution of the peak velocity across the sequence of presentation of the target during control vs. cerebellar block. (**b**) Mean ± 95% confidence intervals of the peak hand velocity for movements to target 1 across all the trial blocks pooled from all four monkeys. For each monkey, the peak hand velocities were normalized to the median peak velocity of the early trials 1–2 from all the control blocks. (**c**) Same as in (**b**) but for targets 2–4. (**d**) Post hoc comparisons between early trials 1–2 and late trials 11–20 during control and cerebellar block. Statistical significance is denoted as follows: p≥0.05 NS, p<0.05*, p<0.01**, p<0.001***. [E: Early, L: Late, C: Control, Cbl: Cerebellar block; T1-8: Targets 1–8].

Over and above this, there was a further progressive decline of the hand velocity during successive movements, most likely due to an adaptive response to the inability to compensate for coupling torques. Furthermore, this adaptive response was not observed in the case of movements where the coupling torques were low, and therefore inter-joint interactions did not play a significant role.

# Slow movements during the cerebellar block cannot fully account for the impaired motor timing

Individuals with cerebellar deficits often exhibit impaired motor timing, leading to movements that are poorly coordinated. Normally, smooth execution of multi-joint movements relies on precise temporal synchronization across joints. However, cerebellar lesions disrupt this temporal coordination, resulting in asynchronous, sequential joint activation resulting in movement decomposition as described earlier (*Bastian et al., 1996*). Moreover, the disrupted temporal control of joint activation introduces variability from trial to trial, causing increased spatial variability in the resulting movement trajectories (*Day et al., 1998*). Thus, movement decomposition and trajectory variability can both be viewed as direct consequences of impaired temporal coordination due to cerebellar dysfunction. These properties of movement are also velocity-dependent (*Bastian et al., 1996*; *Osu et al., 2015*). Therefore, it is conceivable that the combination of primary deficit and adaptive reduction in velocity to reduce the effects of coupling torque observed in our results is the main driver of abnormal motor timing. To evaluate this, we examined movement decomposition and trajectory variability during the outward reaching movements (which exhibited a significant reduction in hand velocity during cerebellar block) and measured the impact of hand velocity on these measures.

To quantify movement decomposition, we normalized the movement times of individual trials to range from 0 to 1 and identified the bins where either the shoulder or the elbow joint paused (speed <20°/s) while the other joint continued moving (*Bastian et al., 1996*). In control trials, decomposition was primarily observed during the early phase of movement, where joint velocities are low making this measure more sensitive (*Figure 5a*, *Figure 5—figure supplement 1a*). However, during cerebellar block trials, decomposition occurred throughout the movement (*Figure 5b* and *Figure 5—figure supplement 1b*) similar to individuals with cerebellar deficits (*Bastian et al., 1996*). To quantify this effect, we calculated a decomposition index for each trial, representing the proportion of movement time during which the movement was decomposed, as defined above. Overall, the cerebellar block increased movement decomposition by 53.5% (CI [42.2, 66.0], p<0.001). Next, we asked whether movement decomposition was mainly due to lower hand velocities. We, therefore, selected a subset of control trials that matched the cerebellar block trials in their peak velocities. Movement decomposition in these control trials was significantly lower than velocity-matched cerebellar block trials (p<0.001; *Figure 5c*; see *Figure 5—figure supplement 1c* for target-wise effects), even though these control trials themselves had 11.0% (CI [5.2, 17.0], p=0.03) higher decomposition than all control trials together. This indicates that even at matched velocities, movements during cerebellar block were more decomposed relative to the control.

Next, we assessed the effect of cerebellar block and hand velocity on trajectory variability. To this end, we first aligned the hand trajectories to have the same starting position and then rotated them so that their endpoint was located on the positive Y-axis (*Figure 5d*; for visualization in joint position space see *Figure 5—figure supplement 2a and b*). Subsequently, we defined the error in each trajectory as the maximal perpendicular distance from the Y-axis. The standard deviation of the errors across all trials was used to quantify the trial-to-trial trajectory variability. When we compared the subset of velocity matched control and cerebellar block trials, we found that cerebellar block exhibited 34.6% (CI [26.2, 43.2], p<0.001) higher trajectory variability (*Figure 5e*; see *Figure 5—figure supplement 2c* for target-wise effects). Normally, slower movements are also less variable due to the speed-accuracy tradeoff (*Plamondon and Alimi, 1997*). Indeed, the trajectory variability in this subset of slower control trials was 5.5% (CI [0.9, 9.9], p=0.02) lower than that of all control trials. In other words, despite slower movements, cerebellar block increased trajectory variability.

Finally, we quantified how cerebellar block affected movement decomposition across successive trials similar to our analysis of hand velocities in *Figure 4*. The cerebellar block-induced increase in movement decomposition was unaffected by the trial sequence (trial type ×trial sequence interaction effect: p=0.43, *Supplementary file 8* and *Figure 5—figure supplement 3*). This further demonstrates that movement decomposition remained high during cerebellar block despite slower movement speeds.

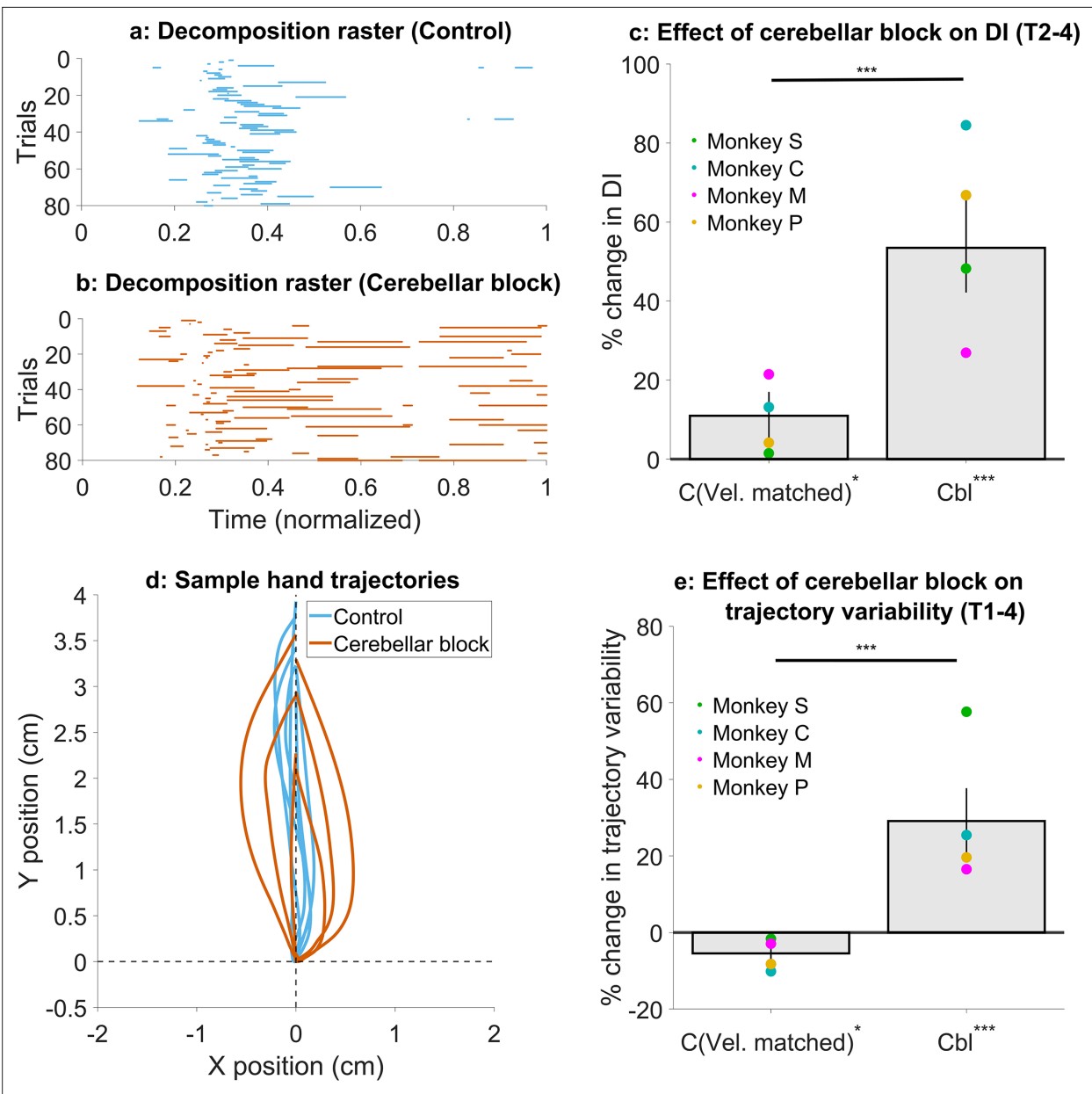

**Figure 5.** Effect of cerebellar block on movement decomposition and trial-to-trial trajectory variability. (**a–c**) Effect of the cerebellar block on the decomposition of movements. (**a**) Binary decomposition raster of sample trials for movements made to target 2 by monkey C during control. The duration of each trial was normalized to 0–1 (bin width = 0.001) and then decomposition was computed for each bin based on whether either (but not both) of the shoulder or the elbow joint velocity was less than 20°/s. (**b**) Same as in (**a**) but for movements to target 2 by monkey C during cerebellar block. (**c**) Change in decomposition index (i.e. the proportion of the movement time during which the movement was decomposed, as defined above) for movements to targets 2–4 during velocity-matched control and cerebellar block trials relative to all control trials. The change in median decomposition was computed for each session. The depicted values are the mean ± 95% confidence intervals across all sessions pooled from all four monkeys. The individual means of each monkey are overlaid. (**d-e**) Effect of the cerebellar block on inter-trial trajectory variability. (**d**) Sample hand trajectories during control vs. cerebellar block trials for movements to target 2 by monkey C. The starting point of the trajectories was shifted to the origin and then they were rotated about the origin so that their endpoints lie on the positive Y-axis. (**e**) Change in inter-trial trajectory variability for movements to targets 2–4 during velocity-matched control and cerebellar block trials relative to all control trials. The trajectory variability was measured as the standard deviation of the maximum perpendicular distance of the trajectories from the Y-axis after transforming them as in (**d**). The change in trajectory variability for the cerebellar block trials was computed relative to the control trials for each session. The depicted values are the mean ± 95% confidence intervals across all sessions pooled from all four monkeys. The individual means of each monkey are overlaid. Statistical significance is denoted as follows: $p \geq 0.05$ NS, $p < 0.05^*$, $p < 0.01^{**}$, $p < 0.001^{***}$. [C: Control, Cbl: Cerebellar block; T1-8: Targets 1–8; DI: Decomposition index].

The online version of this article includes the following figure supplement(s) for figure 5:

*Figure 5 continued on next page*

*Figure 5 continued*

**Figure supplement 1.** Effect of the cerebellar block on the decomposition of movements.

**Figure supplement 2.** Effect of the cerebellar block on the trajectory variability.

**Figure supplement 3.** Effect of the cerebellar block on decomposition index across successive trials for movements to targets 2–4.

## Discussion

Individuals with cerebellar deficits exhibit altered motor behavior related mostly to poor timing and coordination of voluntary movements. Understanding the impairments that follow cerebellar lesions is often complicated by the fact that the studies of this question mostly rely on a heterogeneous population due to various levels and types of cerebellar pathologies. Here, we addressed this question by using high-frequency stimulation to reversibly block cerebellar output to the motor cortex in behaving monkeys wearing an exoskeleton (*Nashef et al., 2019*) and comparing the post-block motor behavior to control conditions. Our results indicate that cerebellar block leads to decreased hand velocity, particularly during the outward reaching movements via two mechanisms: a reduction in muscle torques suggesting motor weakness, and spatially tuned decreases in velocity in targets where large inter-joint interactions occur, indicating a failure to compensate for limb dynamics. Analyzing the time course of these impairments revealed partially overlapping processes: an initial reduction in hand velocity due to the inability to generate sufficient muscle torques, followed by a further progressive decline in hand velocities during subsequent movements during the cerebellar block. This secondary decline was observed only in the case of movements with high inter-joint interactions, suggesting an adaptive strategy to minimize these interactions. Finally, we observed increased motor noise in terms of movement decomposition and trial-to-trial trajectory variability which were independent of the slower movements observed during the cerebellar block.

### Target-dependent reduction in hand velocity and coupling torques during reaching movements during the cerebellar block

Cerebellar block led to a target-specific reduction in hand velocity, even though the arm was fully supported by the exoskeleton and thus the limb weight did not play any role. This observation replicates findings made in previous studies that have documented the slowing of movements in individuals with cerebellar lesions (*Wild and Dichgans, 1993*; *Topka et al., 1998b*, *Konczak et al., 2005*) as well as in monkeys that underwent dentate cooling (*Tsujimoto et al., 1993*). It was argued that the reduction in hand velocity may be a compensatory strategy to counteract the impaired ability to manage inter-joint coupling torques in the absence of cerebellar input to the motor cortex (*Wild and Dichgans, 1993*; *Bastian and Thach, 1995*; *Bastian et al., 1996*; *Beer et al., 2000*). Since coupling torques in multi-joint movements are generated in a velocity-dependent manner (*Hollerbach and Flash, 1982*; *Virji-Babul and Cooke, 1995*), slowing down the movement not only simplifies its kinetics (by reducing the impact of coupling torques) but also allows more time for slower visual feedback to aid in guiding the movement (*Jeannerod, 1988*). Our findings, which demonstrate a correlation between the extent of hand velocity reduction during cerebellar block and the magnitude of coupling torque under control conditions, further corroborate this hypothesis and confirm our approach for reversible inactivation of cerebellar output as a model for ataxia. Furthermore, our model enabled us to study this effect rigorously and systematically while dissociating it from the other primary effects of cerebellar block as discussed below.

### Muscle torque deficit is an acute response to the cerebellar block

Poor compensation for coupling torque is a fundamental deficit in cerebellar ataxia (*Bastian et al., 2000*). However, early studies have reported the occurrence of muscle weakness (asthenia) and hypotonia acutely following cerebellar injury in humans (*Haines et al., 2007*) or experimental lesions in animals (*James et al., 1893*; *Bremer et al., 1935*; *Fulton and Dow, 1937*; *Granit et al., 1955*). Both Luciani (*James et al., 1893*) and *Holmes, 1917* described a triad of cerebellar deficits to be atonia (low muscle tone), asthenia (weakness of voluntary movements), and astasia (oscillation of the head and trunk). Importantly, muscle weakness is typically characteristic of the acute phase of injury, especially in the upper extremity, and tends to normalize gradually over weeks to months, depending on the severity of the injury (*Konczak et al., 2010*). It is, therefore, conceivable that studies in humans

with cerebellar deficits, which often included subjects at the chronic stage of the lesion, may have underestimated the contribution of muscle weakness which by that time was already less pronounced.

We explored the acute outcome of the cerebellar block, a phase in which agonist muscle weakness may play a more prominent role. Importantly, we showed that deficits in the self-generated muscle torque appear immediately after the cerebellar block, resulting in lower hand velocity already in the first few trials of movements. This kind of fast response is less likely to be a strategic response aimed at mitigating the loss of cerebellar signals. This finding is consistent with the initial weakness followed by a gradual recovery of function observed in acute lesions of other neural structures involved in movement control, including the primary (*Pressman and Rosen, 2015*) and premotor cortices (*Freund, 1985*; *Freund and Hummelsheim, 1985*). It also aligns with the understanding that cerebellar output provides a strong excitatory drive to the motor cortex (*Shinoda et al., 1982*; *Hore and Flament, 1988*; *Nashef et al., 2018a*) and several subcortical descending motor systems (*Ruigrok, 2013*), beyond its well-known role in motor timing and coordination. Following the initial reduction in hand velocity observed in our study, there was a further gradual decline across subsequent trials specifically for the movements with high coupling torques, a behavior profile that is more consistent with strategic effort, most likely in response to the inability to compensate for limb dynamics.

The muscle torque, representing the sum of all muscle forces acting at a joint, can also be reduced by co-contraction of agonist and antagonist muscles rather than by agonist weakness alone. However, in cerebellar ataxia, co-contraction has been proposed as a strategy to stabilize stationary joints during decomposed multi-joint movements (*Bastian et al., 1996*). In our experiments, this strategy would likely emerge gradually following cerebellar block similar to the adaptive slowing of movements aimed at reducing inter-joint interactions. We, therefore, argue that the observed acute deficiency in muscle torque was mainly driven by agonist weakness. Our argument is further supported by previous studies which attributed reduced agonist muscle activity as a cause for slowing of voluntary movements in individuals with cerebellar lesions (*Hallett et al., 1991*; *Wild et al., 1996*).

In summary, the slower movements in the absence of cerebellar signals are likely due to two main factors: (i) a primary inability to generate sufficient muscle torques due to the lack of synchronized recruitment of the cortical and sub-cortical descending motor systems via the cerebellar output, and (ii) a secondary adaptive strategy aimed at compensating for the impaired prediction of limb dynamics due to the loss of signals from the cerebellum.

## Insufficient muscle torques vs. poor inter-joint coordination

Unlike most previous clinical studies that examined motor deficits in individuals with chronic cerebellar injuries, our study utilized a reversible model of cerebellar block. This approach enabled us to track the temporal evolution of motor deficits from the onset of the block within the same study participant. Although several studies have explored the nature of motor deficits following cerebellar injuries, there remains considerable debate regarding the fundamental mechanisms driving these deficits. Some studies have suggested that the inability to compensate for inter-joint interactions is the primary driver of the observed motor abnormalities in cerebellar injury (*Bastian et al., 1996*; *Bastian et al., 2000*). In contrast, other studies have emphasized the inability to generate sufficient muscle torques as the core deficit in cerebellar ataxia while claiming that inter-joint coordination is unaffected following cerebellar lesions (*Topka et al., 1998a*, *Boose et al., 1999*). Given these conflicting findings, our study aimed to systematically explore the entire spectrum of motor deficits using a model of reversible cerebellar block. First, we found that the deficits in muscle torque are more pronounced for the reaching movements as compared to the retrieval movements. We identified both an acute insufficiency in muscle torque generation and the inability to compensate for coupling torques as potential mechanisms underlying the observed slowness during the reaching movements. Moreover, we delineated the time course of these deficits, providing insights into their evolution from the onset of cerebellar block.

## The asymmetric effect of cerebellar block on outward vs. inward reaching movements

We observed a notable distinction in the impact of cerebellar blockage on outward vs. inward reaching movements. At a collective level, this variance could be attributed to the disparity in magnitude of coupling torque impulse, with movements towards targets 2–4 exhibiting 18.7% (CI [13.1, 24.1],

p=0.008) higher coupling torque impulse compared to those towards targets 6–8. However, this explanation does not hold consistently when examined at the individual target level. For instance, while the coupling torque was higher for movements towards target 7, the effects of cerebellar blockage were less pronounced compared to those observed for target 1, where the coupling torques are considerably lower. There are two possible explanations for this discrepancy. One possibility is that the effect of cerebellar block exhibits a bias in the direction of shoulder flexion (i.e. for movements to targets 1–4). Several early studies in cats (*Hare et al., 1936*; *Chambers, 1947*), non-human primates (*Magoun et al., 1935*), and humans *Nashold and Slaughter, 1969* have all reported a bias towards the activation of ipsilateral flexors of the arm following stimulation of the deep cerebellar nuclei and the superior cerebellar peduncle. Furthermore, lesioning of these structures led to the reduction of ipsilateral flexor tone and power (*Schneider and Crosby, 1963*; *Nashold and Slaughter, 1969*). The cerebellar output influences the excitability of spinal interneurons either directly (*Bantli and Bloedel, 1975*; *Asanuma et al., 1980*; *Sathyamurthy et al., 2020*) or via indirect relays in the subcortical descending motor systems like the rubrospinal and the reticulospinal pathways (*Ruigrok, 2013*). Since the spinal interneurons influence the activity of flexor muscles twice as often as extensor muscles (*Perlmutter et al., 1998*), this might explain why the cerebellar block had a stronger effect on movements involving shoulder flexion (i.e. outward reaching movements) relative to shoulder extension (i.e. inward reaching movements) in our study. Alternatively, the initial position of the limb in our experimental setup (i.e. on the central target) might have been favorable for movements in the direction of shoulder extension, thereby making them easier to perform.

### Asynchronous joint movements and motor variability occur despite slower movements in cerebellar block

Abnormally high motor variability (*Day et al., 1998*; *Timmann et al., 1999*; *Schlerf et al., 2013*) and decomposition of multi-joint movements (*Bastian et al., 1996*) are core deficits in cerebellar patients. We showed that the cerebellar block decomposed the dual-joint movements into temporally isolated single-joint actions and led to an increased trial-to-trial spatial noise. These two measures can be affected by the reduction in movement velocities. Our results showed that even though movement decomposition increased with a reduction in velocity in control trials, they were still significantly lower than that of velocity-matched cerebellar block trials. Second, the trial-to-trial spatial noise in hand trajectories increased during cerebellar block despite the slower movement speeds. The cerebellum is known to participate in the planning of movements by providing an internal model of the limb which is necessary for predicting the sensory outcomes of motor commands (*Wolpert et al., 1998*). For this purpose, the cerebellum aids in estimating the state of the limb by integrating the sensory information of its last known state with predictions of its response to the latest motor command (*Miall et al., 2007*; *Miall and King, 2008*). Therefore, in the absence of cerebellar signals, there are programming errors in the motor command required to launch the limb accurately toward a target (*Day et al., 1998*). These errors may in turn translate into the noisy joint activation patterns and variable hand trajectories observed in our results. Moreover, the persistence of these deficits independently of the movement speeds observed during cerebellar block indicates that they cannot be compensated for, unlike the impaired prediction of inter-joint interactions which are strategically dealt with by reducing the movement velocity.

In summary, our study systematically investigated the temporal progression and distinct mechanisms of motor deficits in a controlled, reversible animal model of cerebellar block. These include an acute onset muscle torque deficit, impaired compensation of limb dynamics, and increased motor noise. We aim to further explore the neural correlates of these deficits in future studies.

## Materials and methods
### Experimental subjects

This study was performed on four adult female monkeys (*Macaca fascicularis*, wt 4.5–8 kg). The care and surgical procedures of the subjects was in accordance with the Hebrew University Guidelines for the Use and Care of Laboratory Animals in Research, supervised by the Institutional Committee for Animal Care and Use (ethics approval numbers MD13135894 and MD19158354).

## Behavioral task

Each monkey was trained to sit in a primate chair and perform planar, center-out movements. During the task, the left upper arm was abducted 90° and rested on cushioned troughs secured to links of a two-joint exoskeleton (KINARM, BKIN Technologies Ltd.). The sequence of events in each trial of this task was as follows. First, the monkey located a cursor (projected on a horizontal screen immediately above its arm) within a central target (2–3 cm in diameter). After 500–800 ms, a randomly selected peripheral target (out of 8 evenly distributed targets) was displayed. The monkey had to wait 450–850 ms until the central target disappeared (i.e. the 'go signal) and then reached the cued peripheral target (2–3 cm in diameter) located 4–5 cm away. For monkeys C and M, we inserted a 200 ms grace period before the 'go' signal to encourage predictive timing. The onset of movement within this time before the 'go' signal did not abort the trial. We limited the time to reach the peripheral target after the 'go' signal to 500 ms for monkeys M and C and 800–900 ms for monkeys S and P. Following successful reach, the monkeys were rewarded with a drop of applesauce. It is important to note that the monkeys were required to hold at the target briefly—100 ms for Monkeys S and P, and 150 ms for Monkeys C and M—before receiving the reward. However, given the size of the targets and the velocity of movements, it often happened that the monkeys didn't have to stop their movements fully to obtain the reward. Importantly, we relaxed the task's requirements (by increasing the target size and reducing the temporal constraints) to enable the monkeys to perform a sufficient number of successful trials under both the control and the cerebellar block conditions. This was necessary as we found that strict criteria for these parameters yielded a very low success rate in the cerebellar block condition. Given these practical constraints, we note that this task design is likely suboptimal for studying endpoint accuracy which is an important aspect of cerebellar control (*Low et al., 2018*; *Becker and Person, 2019*; *Calame et al., 2023*).

## Insertion of the stimulating electrode into the superior cerebellar peduncle

After the training was completed, a square recording chamber (24×24 mm) was attached to the monkey's skull above the upper limb-related area of the motor cortex in a surgical procedure under general anesthesia. To insert a chronic stimulating electrode into the ipsilateral superior cerebellar peduncle (SCP), a round chamber (diameter = 19 mm) was also implanted above the estimated insertion point (as per stereotactic measurements) during the surgery for the cortical chamber implantation. Subsequently, a post-surgery MRI was used to plan the trajectory for the electrode insertion into the SCP. After a recovery and re-training period, a bipolar concentric electrode (Microprobes for Life Science Inc) was inserted through the round chamber along the planned trajectory. During the process of insertion, the evoked multi-unit activity in the primary motor cortex was simultaneously monitored to identify the precise location where the stimulating electrode should be secured (*Nashef et al., 2018a*; *Nashef et al., 2018b*).

## High-frequency stimulation to block the cerebellar outflow to the motor cortex

The superior cerebellar peduncle was blocked using high-frequency stimulation (HFS). This stimulation protocol consisted of a train of biphasic rectangular-pulse stimuli (where each phase was 200 µs) applied at 130 Hz. Each train was delivered at a fixed intensity of 150–250 µA. In our earlier study, we demonstrated that by using this manipulation, we can successfully replicate the impaired motor timing and coordination characteristic of cerebellar ataxia (*Nashef et al., 2019*). Furthermore, we also demonstrated that at the motor cortical level, this manipulation leads to a loss of response transients at movement onset and decoupling of task-related activity, similar to findings observed in studies where the dentate nucleus was cooled (*Meyer-Lohmann et al., 1975*). Deep brain stimulation (DBS) is a perturbation similar in properties to the HFS we used, which is routinely used in clinical practice for similar purposes of interfering with neural circuitries that underwent pathological changes (*Perlmutter and Mink, 2006*) although in both applications there is no accurate measure for the extent of the block (i.e. the fraction of fibers which become ineffective in these conditions) other than the behavioral consequences. It is possible that HFS probably imposes only a partial block of the CTC system.

It is important to note that the superior cerebellar peduncle is known to carry input and output fibers to and from several neural structures (*Naidich et al., 2013*). Therefore, we cannot rule out the fact that the effect of HFS may be mediated in part through pathways other than the cerebello-thalamo-cortical pathway (as mentioned in the Discussion section). However, in primates, the cerebellar-thalamo-cortical pathway greatly expanded (at the expense of the cerbello-rubro-spinal tract) in mediating cerebellar control of voluntary movements (*Horne and Butler, 1995*). The cerebello-subcortical pathways diminished in importance throughout evolution (*Padel et al., 1981*; *Nathan and Smith, 1982*; *ten Donkelaar, 1988*). Previously, we found that the ascending spinocerebellar axons which enter the cerebellum through the superior cerebellar peduncle (SCP) are weakly task-related and the descending system is quite small (*Cohen et al., 2017*). Nevertheless, we acknowledge that HFS disrupts cerebellar communication to the rest of the nervous system broadly, rather than solely the cerebello-thalamo-cortical pathway.

## Data collection protocol

Each monkey performed the experimental task for several days (sessions). A total of 46, 29, 36, and 54 sessions of data were collected from Monkeys S, C, M, and P, respectively. Each session was divided into three or four sub-sessions. Each sub-session included blocks of control trials (~80 trials) and trials with the superior cerebellar peduncle blocked using high-frequency stimulation (~50 trials) as described above. During the task performance, the following behavioral data was obtained at 1000 Hz using the KINARM's motor encoders: (i) Hand position in the x-y plane of movement, (ii) Elbow and shoulder joint angular positions, velocities, and accelerations. The lengths of the upper arm and forearm of each monkey were measured as the distances from the head of the humerus to the lateral epicondyle and from the lateral epicondyle to the palm, respectively. The inertial properties of the upper limb were estimated using the monkey's body weight (*Cheng and Scott, 2000*).

## Processing of the behavioral data

All data analysis was performed in MATLAB R2021b (MathWorks Inc). The raw kinematic data was low-pass filtered by using a second order Butterworth filter (cutoff = 10 Hz) and then epoched into trials. For this purpose, the start and end events were taken as the time of presentation of the cue (i.e. the peripheral target) and the time of entry of the control cursor into the peripheral target, respectively. This cutoff meant that a major portion of the extracted segment of each trial was the acceleration phase of the movement. The movement onset was defined as the time when the radial hand velocity exceeded 5% of its peak. Subsequently, all the trials were zero-centered around the calculated movement onset. We then applied a two-link inverse dynamics model of the upper arm to the kinematic data to compute the coupling and muscle torques acting on the elbow and shoulder joints (*Fagg et al., 2009*). The inverse dynamics equations of motion for the combined system consisting of the monkey's arm and the associated moving components of the KINARM were used to compute the joint torques:

$$T_{me} = T_{ne} - T_{ce} \tag{i}$$
$$T_{ms} = T_{ns} - T_{cs} \tag{ii}$$

where $T_{ne}$ and $T_{ns}$ are the net torques (proportional to the joint's acceleration), $T_{ce}$ and $T_{cs}$ are passive coupling torques, and $T_{me}$, $T_{ms}$ are the active muscle torques acting at the elbow and shoulder joints, respectively. The expanded formulas for computing each of the above terms are provided by *equations 1a-2*e in the methods section C (*Fagg et al., 2009*). It must be noted that the 'muscle' torques computed here include the effects of the actively generated muscle forces as well as the viscoelastic effects that result from the musculoskeletal system and the KINARM (*Smith and Zernicke, 1987*). Data from some of the same animals was presented in prior work where we showed that during reaching, cerebellar block resulted in reduced reach velocity, increased trajectory curvature, and shoulder-elbow dyscoordination (*Nashef et al., 2019*). In this study, we extend these findings by using inverse-dynamics modeling to explore the relationship between altered kinematics and joint torques. Furthermore, the analyses exploring progressive velocity changes across trials under the cerebellar block and the relationship of motor noise to movement velocity are newly reported in this study.

To strictly focus on feedforward control, we could have examined the measured variables in the first 50–100 ms of the movement which has been shown to be unaffected by feedback responses (*Todorov*

*and Jordan, 2002*; *Pruszynski et al., 2008*; *Pruszynski and Scott, 2012*; *Crevecoeur and Scott,* *2013*). However, in our task, the amplitude of movements made by our monkeys was small, and therefore the response measures we used were too small in the first 50–100 ms for a robust estimation. Also, fixing a time window led to an unfair comparison between control and cerebellar block trials, in which velocity was significantly reduced and, therefore, movement time was longer. Therefore, we used the peak velocity, torque-impulse at the peak velocity, and maximum deviation of the hand trajectory as response measures.

## Statistical analysis

We used linear mixed effects analysis to evaluate the effects of predictor variables of interest on the computed response variables. This allowed us to account for the variability across the monkeys which was modeled as a random effect. Each data sample for this model was obtained by calculating the median of all the trials for each unique combination of the response variable(s) per session. We computed the change in the measured outcome variable due to the cerebellar block per session using the following formula:

$$\frac{\textit{Median response during cerebellar block} - \textit{median response during control}}{\textit{Median response during control}} \times 100$$

For post-hoc comparisons, we used the 'coefTest' function from MATLAB's 'fitlme' to perform hypothesis testing on the fixed effects. This involved specifying the contrast matrices to evaluate specific hypotheses regarding the differences in the response variables across the levels of the predictor variables. The 'coefTest' function allowed us to obtain p-values for these comparisons. The computed p-values were then corrected using the Benjamini-Hochberg procedure to control the false discovery rate for multiple post-hoc comparisons, wherever applicable (*Haynes, 2013*). A significance level of 0.05 was used for all comparisons in this study. For the estimation of the group means and their confidence intervals, we pooled the per session data from all monkeys after ensuring that effects are consistent across the monkeys. For further clarity, we have also overlayed the means of the measured variables from individual monkeys on all our plots of summary statistics.

## Acknowledgements

This work was funded by the Israel Science Foundation (ISF-1801/18 and ISF 1207/23) to YP, the Deutsche Forschungsgemeinschaft (DFG, German Research Foundation Project-ID 431549029–SFB 1451) to YP, National Institute of Neurological Disorders and Stroke of the National Institutes of Health (NIH R01NS110901) to YP, the National Institutes of Health (NIH R01NS105759) to JPAD, the Council for Higher Education PhD Sandwich Fellowship, Government of Israel to NS for undertaking his dual-PhD at the Hebrew University and Northwestern University, and the generous support of the Baruch Foundation to YP. The authors would like to thank Dr. Abdulraheem Nashef, a postdoctoral fellow at the University of Colorado for collecting part of the data used in this study.

## Additional information

### Funding

| Funder | Grant reference number | Author |
| --- | --- | --- |
| Israel Science Foundation | ISF-1801/18 | Yifat Prut |
| Israel Science Foundation | ISF 1207/23 | Yifat Prut |
| Deutsche Forschungsgemeinschaft | 431549029-SFB 1451 | Yifat Prut |
| National Institute of Neurological Disorders and Stroke | NIH R01NS110901 | Yifat Prut |

| Funder | Grant reference number | Author |
| --- | --- | --- |
| National Institutes of Health | R01NS105759 | Julius PA Dewald |
| Council for Higher Education | PhD Sandwich Fellowship | Nirvik Sinha |
| Baruch Foundation | Financial support | Yifat Prut |

The funders had no role in study design, data collection and interpretation, or the decision to submit the work for publication.

## Author contributions

Nirvik Sinha, Conceptualization, Data curation, Software, Formal analysis, Investigation, Visualization, Methodology, Writing – original draft, Writing – review and editing; Sharon Israely, Formal analysis, Investigation, Methodology; Ora Ben Harosh, Ran Harel, Investigation, Methodology; Julius PA Dewald, Conceptualization, Supervision, Funding acquisition, Writing – original draft, Writing – review and editing; Yifat Prut, Conceptualization, Software, Supervision, Funding acquisition, Investigation, Methodology, Writing – original draft, Project administration, Writing – review and editing

## Author ORCIDs

Nirvik Sinha ⓘ https://orcid.org/0000-0001-6960-7159
Sharon Israely ⓘ https://orcid.org/0000-0003-4298-7395
Ora Ben Harosh ⓘ http://orcid.org/0009-0009-2587-834X
Ran Harel ⓘ https://orcid.org/0000-0002-0165-7114
Julius PA Dewald ⓘ https://orcid.org/0000-0003-0641-8400
Yifat Prut ⓘ https://orcid.org/0000-0003-1988-8794

## Ethics

This study was performed on four adult female monkeys (Macaca fascicularis, weight 4.5-8 kg). The care and surgical procedures of the subjects will be in accordance with the Hebrew University Guidelines for the Use and Care of Laboratory Animals in Research, supervised by the Institutional Committee for Animal Care and Use (ethics approval numbers MD13135894 and MD19158354).

Reviewer #1 (Public review): https://doi.org/10.7554/eLife.105152.4.sa1
Reviewer #2 (Public review): https://doi.org/10.7554/eLife.105152.4.sa2
Reviewer #3 (Public review): https://doi.org/10.7554/eLife.105152.4.sa3
Author response https://doi.org/10.7554/eLife.105152.4.sa4

# Additional files

## Supplementary files

Supplementary file 1. Mean success rate across sessions per monkey.

Supplementary file 2. ANOVA marginal tests for the effect of target direction on the change in peak hand velocity due to cerebellar block relative to control (DF: degrees of freedom). Movements exhibited a significant reduction in peak hand velocity during the cerebellar block in a target-dependent manner. The change in peak hand velocity was modeled using a linear mixed-effects model, with target as a fixed effect and random intercepts and slopes for target within each subject (i.e. monkey). For each session, the target-wise change in the median peak hand velocity during the cerebellar block trials was computed relative to that of control trials. The input to the model was the target-wise values computed from all sessions pooled across all four monkeys. The significant effect of target direction on the change in peak velocity can be interpreted as analogous to the interaction between cerebellar block and target direction on the actual peak velocities.

Supplementary file 3. Comparison of peak hand velocities between outward and inward reaching movements during control conditions. (a) ANOVA marginal tests for the effect of target group (targets 1–4 vs. targets 5–8) on peak hand velocity during control (DF: degrees of freedom). Movements exhibited no significant difference in peak hand velocity during the outward reaching (targets 1–4) vs. retrieval (targets 5–8) movements during the control condition. The peak hand velocity was modeled using a linear mixed-effects model, with the target group (T1-4 vs. T5-8)

as a fixed effect and random intercepts and slopes for the target group within each subject (i.e. monkey). The median peak hand velocity during the control trials was computed for each session. The input to the model was the target group-wise values computed from all sessions pooled across all four monkeys. (b) Mean peak hand velocity for outward reaching (targets 1–4) vs. inward reaching (targets 5–8) movements during control across all sessions per monkey. For each session, the median peak hand velocity was computed across all the control trials per target group. Then, the mean and confidence intervals of the mean were computed from the per-session data for each monkey.

Supplementary file 4. ANOVA marginal tests for the effect of target direction on the change in shoulder muscle torque impulse due to cerebellar block relative to control (DF: degrees of freedom). Movements exhibited a significant reduction in shoulder muscle torque impulse during the cerebellar block in a target-dependent manner. The torque impulse was computed by integrating the torque profile during the positive acceleration phase of the movement. The change in muscle torque impulse was modeled using a linear mixed-effects model, with target as a fixed effect and random intercepts and slopes for target within each subject (i.e. monkey). For each session, the target-wise change in the median muscle torque impulse during the cerebellar block trials was computed relative to that of control trials. The input to the model was the target-wise values computed from all sessions pooled across all four monkeys. The significant effect of target direction on the change in muscle torque impulse can be interpreted as analogous to the interaction between cerebellar block and target direction on the actual peak velocities.

Supplementary file 5. ANOVA marginal tests for the effect of target direction on the change in elbow muscle torque impulse due to cerebellar block relative to control (DF: degrees of freedom). Movements exhibited a significant reduction in elbow muscle torque impulse during the cerebellar block in a target-dependent manner. The torque impulse was computed by integrating the torque profile during the positive acceleration phase of the movement. The change in muscle torque impulse was modeled using a linear mixed-effects model, with target as a fixed effect and random intercepts and slopes for target within each subject (i.e. monkey). For each session, the target-wise change in the median muscle torque impulse during the cerebellar block trials was computed relative to that of control trials. The input to the model was the target-wise values computed from all sessions pooled across all four monkeys.

Supplementary file 6. ANOVA marginal tests for the effect of trial sequence and trial type (control/cerebellar block) on the peak hand velocity relative to first two trials in control for movements to target 1 (DF: degrees of freedom). The evolution of peak hand velocities during control vs. cerebellar block was analyzed by preserving the order of presentation of the target in each block of trials (i.e. trial sequence). For each monkey, the peak hand velocities were normalized to the median peak velocity of the early trials 1–2 in the control blocks. The normalized peak velocities were then modeled using a linear mixed-effects model, with trial type (control/cerebellar block) and trial sequence (1-20) as fixed effects and random intercepts and slopes for trial type and trial sequence within each subject (i.e. monkey).

Supplementary file 7. ANOVA marginal tests for the effect of trial sequence and trial type (control/cerebellar block) on the peak hand velocity relative to first two trials in control for movements to target 2–4 (DF: degrees of freedom). The evolution of peak hand velocities during control vs. cerebellar block was analyzed by preserving the order of presentation of the target in each block of trials (i.e. trial sequence). For each monkey, the peak hand velocities were normalized to the median peak velocity of the early trials 1–2 in the control blocks. The normalized peak velocities were then modeled using a linear mixed-effects model, with trial type (control/cerebellar block) and trial sequence (1-20) as fixed effects and random intercepts and slopes for trial type and trial sequence within each subject (i.e. monkey).

Supplementary file 8. ANOVA marginal tests for the effect of trial sequence and trial type (control/cerebellar block) on movement decomposition relative to first two trials in control for movements to targets 2–4 (DF: degrees of freedom). The evolution of decomposition index (measured as the fraction of time during a movement when either, but not both, of the shoulder or the elbow joint velocity was less than 20°/s) during control vs. cerebellar block was analyzed by preserving the order of presentation of the target in each block of trials (i.e. trial sequence). For each monkey, the decomposition indices were normalized to the median decomposition index of the early trials 1–2 in the control blocks. The normalized decomposition indices were then modeled using a linear mixed-effects model, with trial type (control/cerebellar block) and trial sequence (1-20) as fixed effects and random intercepts and slopes for trial type and trial sequence within each subject (i.e. monkey).

MDAR checklist

## Data availability

Data and code used in this manuscript are available publicly as a GitHub repository: https://github.com/NirvikNU/Disentangling-acute-motor-deficits-and-adaptive-responses-evoked-by-the-loss-of-cerebellar-output- (copy archived at *Sinha, 2025*). The matlab fig files used in this publication is available in the 'plots' folder. The data used in the individual plots can be extracted from the .fig files. Please contact corresponding authors for further requests.

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
