## [Editor Report · eLife Assessment]

Using a unique cerebellar disruption approach in non-human primates, this study provides **valuable** new insight into how cerebellar inputs to the motor cortex contribute to reaching. The findings **convincingly** demonstrate that reaching movements following cerebellar disruption slow down because of both an acute deficit in producing muscle activity as well as a progressive decline in compensating for limb dynamics. This work will be of interest to neuroscientists and clinicians interested in cerebellar function and pathology.

---

## [Referee Report · Reviewer #1 (Public review)]

Summary:

In a previous work Prut and colleagues had shown that during reaching, high frequency stimulation of the cerebellar outputs resulted in reduced reach velocity. Moreover, they showed that the stimulation produced reaches that deviated from a straight line, with the shoulder and elbow movements becoming less coordinated. In this report they extend their previous work by addition of modeling results that investigate the relationship between the kinematic changes and torques produced at the joints. The results show that the slowing is not due to reductions in interaction torques alone, as the reductions in velocity occur even for movements that are single joint. More interestingly, the experiment revealed evidence for decomposition of the reaching movement, as well as an increase in the variance of the trajectory.

Strengths:

This is a rare experiment in a non-human primate that assessed the importance of cerebellar input to the motor cortex during reaching.

Weaknesses:

None

---

## [Referee Report · Reviewer #2 (Public review)]

This manuscript asks an interesting and important question: what part of 'cerebellar' motor dysfunction is an acute control problem vs a compensatory strategy to the acute control issue? The authors use a cerebellar 'blockade' protocol, consisting of high frequency stimuli applied to the cerebellar peduncle which is thought to interfere with outflow signals. This protocol was applied in monkeys performing center out reaching movements and has been published from this laboratory in several preceding studies. I found the take-home-message broadly convincing and clarifying - that cerebellar block reduces muscle activation acutely particularly in movements that involve multiple joints and therefore invoke interaction torques, and that movements progressively slow down to in effect 'compensate' for these acute tone deficits. The manuscript was generally well written, data were clear, convincing and novel. The key strengths are differentiating acute from sub-acute (within session but not immediate) kinematic consequences of cerebellar block.

---

## [Referee Report · Reviewer #3 (Public review)]

Summary:

In their revised manuscript, Sinha and colleagues aim to identify distinct causes of motor impairments seen when perturbing cerebellar circuits. This goal is an important one, given the diversity of movement related phenotypes in patients with cerebellar lesion or injury, which are especially difficult to dissect given the chronic nature of the circuit damage. To address this goal, the authors use high-frequency stimulation (HFS) of the superior cerebellar peduncle in monkeys performing reaching movements. HFS provides an attractive approach for transiently disrupting cerebellar function previously published by this group. First, they find a reduction in hand velocities during reaching, which was more pronounced for outward versus inward movements. By modeling inverse dynamics, they find evidence that shoulder muscle torques are especially affected. Next, the authors examine the temporal evolution of movement phenotypes over successive blocks of HFS trials. Using this analysis, they find that in addition to the acute, specific effects on torques in early HFS trials, there was an additional progressive reduction in velocity during later trials, which they interpret as an adaptive response to the inability to effectively compensate for interaction torques during cerebellar block. Finally, the authors examine movement decomposition and trajectory, finding that even when low velocity reaches are matched to controls, HFS produces abnormally decomposed movements and higher than expected variability in trajectory.

Strengths:

Overall, this work provides important insight into how perturbation of cerebellar circuits can elicit diverse effects on movement across multiple timescales.

The HFS approach provides temporal resolution and enables analysis that would be hard to perform in the context of chronic lesions or slow pharmacological interventions. Thus, this study describes an important advance over prior methods of circuit disruption in the monkey, and their approach can be used as a framework for future studies that delve deeper into how additional aspects of sensorimotor control are disrupted (e.g., response to limb perturbations).

In addition, the authors use well-designed behavioral approaches and analysis methods to distinguish immediate from longer-term adaptive effects of HFS on behavior. Moreover, inverse dynamics modeling provides important insight into how movements with different kinematics and muscle dynamics might be differentially disrupted by cerebellar perturbation.

Remaining comments:

The argument that there are acute and adaptive effects to perturbing cerebellar circuits is compelling, but there seems to be a lost opportunity to leverage the fast and reversible nature of the perturbations to further test this idea and strengthen the interpretation. Specifically, the authors could have bolstered this argument by looking at the effects of terminating HFS - one might hypothesize that the acute impacts on joint torques would quickly return to baseline in the absence of HFS, whereas the longer-term adaptive component would persist in the form of aftereffects during the 'washout' period. As is, the reversible nature of the perturbation seems underutilized in testing the authors' ideas. While this experimental design was not implemented here, it seems like a good opportunity for future work using these approaches.

The analysis showing that there is a gradual reduction in velocity during what the authors call an adaptive phase is convincing. While it is still not entirely clear why disruption of movement during the adaptive phase is not seen for inward targets, despite the fact that many of the inward movements also exhibit large interaction torques, the authors do raise potential explanations in the Discussion.

---

## [Author Response]

The following is the authors’ response to the previous reviews

**Reviewer #1 (Public review):**
Summary:In a previous work Prut and colleagues had shown that during reaching, high frequency stimulation of the cerebellar outputs resulted in reduced reach velocity. Moreover, they showed that the stimulation produced reaches that deviated from a straight line, with the shoulder and elbow movements becoming less coordinated. In this report they extend their previous work by addition of modeling results that investigate the relationship between the kinematic changes and torques produced at the joints. The results show that the slowing is not due to reductions in interaction torques alone, as the reductions in velocity occur even for movements that are single joint. More interestingly, the experiment revealed evidence for decomposition of the reaching movement, as well as an increase in the variance of the trajectory.Strengths:This is a rare experiment in a non-human primate that assessed the importance of cerebellar input to the motor cortex during reaching.Weaknesses:None
**Reviewer #1 (Recommendations for the authors):**
The authors have answered my questions adequately and I have no further comments.
**Reviewer #2 (Public review):**
This manuscript asks an interesting and important question: what part of 'cerebellar' motor dysfunction is an acute control problem vs a compensatory strategy to the acute control issue? The authors use a cerebellar 'blockade' protocol, consisting of high frequency stimuli applied to the cerebellar peduncle which is thought to interfere with outflow signals. This protocol was applied in monkeys performing center out reaching movements and has been published from this laboratory in several preceding studies. I found the takehome-message broadly convincing and clarifying - that cerebellar block reduces muscle activation acutely particularly in movements that involve multiple joints and therefore invoke interaction torques, and that movements progressively slow down to in effect 'compensate' for these acute tone deficits. The manuscript was generally well written, data were clear, convincing and novel. The key strengths are differentiating acute from subacute (within session but not immediate) kinematic consequences of cerebellar block.
**Reviewer #2 (Recommendations for the authors):**
I think the manuscript is good as is. That said, it would have been nice to see more of the behavioral outcomes in Figure 5 (e.g. decomposition and trajectory variability) analyzed longitudinally like the velocity measurements in Fig. 4. This would clearly strengthen the insight into acute and compensatory components of cerebellar motor deficits.

The two behavioral measures of motor noise used in our study are movement decomposition and trajectory variability (Figure 5). Since trajectory variability is measured across trials we could not analyze this measure longitudinally as a function of trial number. However, following the reviewer’s advice, we examined movement

decomposition for successive trials in control vs. cerebellar block for movements to targets 2-4 similar to the analysis of hand velocity in figure 4. We found no interaction effect between trial sequence x cerebellar block on movement decomposition. This result is consistent with our conclusion that noisy joint activation occurs independently of adaptive slowing of multi-joint movements. We have updated our main text (lines 293-299) and supplementary information (supplementary figure S5 and supplementary table S8) to include this result.

**Reviewer #3 (Public review):**
Summary:In their revised manuscript, Sinha and colleagues aim to identify distinct causes of motor impairments seen when perturbing cerebellar circuits. This goal is an important one, given the diversity of movement related phenotypes in patients with cerebellar lesion or injury, which are especially difficult to dissect given the chronic nature of the circuit damage. To address this goal, the authors use high-frequency stimulation (HFS) of the superior cerebellar peduncle in monkeys performing reaching movements. HFS provides an attractive approach for transiently disrupting cerebellar function previously published by this group. First, they find a reduction in hand velocities during reaching, which was more pronounced for outward versus inward movements. By modeling inverse dynamics, they find evidence that shoulder muscle torques are especially affected. Next, the authors examine the temporal evolution of movement phenotypes over successive blocks of HFS trials. Using this analysis, they find that in addition to the acute, specific effects on torques in early HFS trials, there was an additional progressive reduction in velocity during later trials, which they interpret as an adaptive response to the inability to effectively compensate for interaction torques during cerebellar block. Finally, the authors examine movement decomposition and trajectory, finding that even when low velocity reaches are matched to controls, HFS produces abnormally decomposed movements and higher than expected variability in trajectory.Strengths:Overall, this work provides important insight into how perturbation of cerebellar circuits can elicit diverse effects on movement across multiple timescales.The HFS approach provides temporal resolution and enables analysis that would be hard to perform in the context of chronic lesions or slow pharmacological interventions. Thus, this study describes an important advance over prior methods of circuit disruption in the monkey, and their approach can be used as a framework for future studies that delve deeper into how additional aspects of sensorimotor control are disrupted (e.g., response to limb perturbations).In addition, the authors use well-designed behavioral approaches and analysis methods to distinguish immediate from longer-term adaptive effects of HFS on behavior. Moreover, inverse dynamics modeling provides important insight into how movements with different kinematics and muscle dynamics might be differentially disrupted by cerebellar perturbation.In this revised version of the manuscript, the authors have provided additional analyses and clarification that address several of the comments from the original submission.Remaining comments:The argument that there are acute and adaptive effects to perturbing cerebellar circuits is compelling, but there seems to be a lost opportunity to leverage the fast and reversible nature of the perturbations to further test this idea and strengthen the interpretation. Specifically, the authors could have bolstered this argument by looking at the effects of terminating HFS - one might hypothesize that the acute impacts on joint torques would quickly return to baseline in the absence of HFS, whereas the longer-term adaptive component would persist in the form of aftereffects during the 'washout' period. As is, the reversible nature of the perturbation seems underutilized in testing the authors' ideas. While this experimental design was not implemented here, it seems like a good opportunity for future work using these approaches.

We agree with the reviewer that examining the effect of the cerebellar block on immediate post-block washout trials in future studies will be insightful.

The analysis showing that there is a gradual reduction in velocity during what the authors call an adaptive phase is convincing. While it is still not entirely clear why disruption of movement during the adaptive phase is not seen for inward targets, despite the fact that many of the inward movements also exhibit large interaction torques, the authors do raise potential explanations in the Discussion.The text in the Introduction and in the prior work developing the HFS approach overstates the selectivity of the perturbations. First, there is an emphasis on signals transmitted to the neocortex. As the authors state several times in the Discussion, there are many subcortical targets of the cerebellar nuclei as well, and thus it is difficult to disentangle target-specific behavioral effects using this approach. Second, the superior cerebellar peduncle contains both cerebellar outputs and inputs (e.g., spinocerebellar). Therefore, the selectivity in perturbing cerebellar output feels overstated. Readers would benefit from a more agnostic claim that HFS affects cerebellar communication with the rest of the nervous system, which would not affect the major findings of the study. In the revised manuscript, the authors do provide additional anatomical and evolutionary context and discuss potential limitations in the selectivity of HFS in the Materials and Methods. However, I feel that at least a brief mention of these caveats in the Introduction, where it is stated, "we then reversibly blocked cerebellar output to the motor cortex", would benefit the reader.

Following the advice of the reviewer, we have now revised the introduction section of our manuscript in the following way (lines 61-67):

“…We then reversibly disrupted cerebellar communication with other neural structures using high-frequency stimulation (HFS) of the superior cerebellar peduncle, assessing the impact of this perturbation on subsequent movements. Although our approach primarily affects cerebellar output to the motor cortex, it also disrupts fibers carrying input signals (e.g., spinocerebellar) and pathways to various subcortical targets (e.g., cerebellorubrospinal). Thus, our manipulation broadly interferes with cerebellar communication…”

**Reviewer #3 (Recommendations for the authors):**
Typo on line 102; "subs-sessions"

We have corrected this typographical error in our revised manuscript (line 106).